# Contractility-induced self-organization of smooth muscle cells: from multilayer cell sheets to dynamic three-dimensional clusters

Xiuyu Wang [1,4✉], David Gonzalez-Rodriguez [2], Thomas Vourc'h [3,5], Pascal Silberzan [3] & Abdul I. Barakat [1✉]

Smooth muscle cells (SMCs) are mural cells that play a vital contractile function in many tissues. Abnormalities in SMC organization are associated with many diseases including atherosclerosis, asthma, and uterine fibroids. Various studies have reported that SMCs cultured on flat surfaces can spontaneously form three-dimensional clusters whose organization resembles that encountered in some of these pathological settings. Remarkably, how these structures form remains unknown. Here we combine in vitro experiments and physical modeling to show that three-dimensional clusters initiate when cellular contractile forces induce a hole in a flat SMC sheet, a process that can be modeled as the brittle fracture of a viscoelastic material. The subsequent evolution of the nascent cluster can be modeled as an active dewetting process with cluster shape evolution driven by a balance between cluster surface tension, arising from both cell contractility and adhesion, and cluster viscous dissipation. The description of the physical mechanisms governing the spontaneous emergence of these intriguing three-dimensional clusters may offer insight into SMC-related disorders.

[1] LadHyX, CNRS, Ecole Polytechnique, Institut Polytechnique de Paris, Palaiseau, France. [2] Université de Lorraine, LCP-A2MC Metz, France. [3] Laboratoire PhysicoChimie Curie, Institut Curie, PSL Research University, Paris, France. [4] Present address: Laboratoire Matière et Systèmes Complexes (MSC), UMR 7057, CNRS and Université de Paris, 75013 Paris, France. [5] Present address: Université Clermont Auvergne, SIGMA Clermont, Institut Pascal, BP 10448, F-63000 Clermont-Ferrand, France. ✉email: xiuyu.wang.fr@gmail.com; abdul.barakat@polytechnique.edu

Smooth muscle cells (SMCs) are contractile cells that populate the walls of many hollow tissues and organs including blood vessels, pulmonary airways, and the gastrointestinal and urogenital systems[1,2]. In vivo, SMCs regulate tissue tone and synthesize a host of extracellular matrix proteins. Aberrant SMC proliferation and organization are the hallmarks of many chronic pathologies including atherosclerosis[3], hypertension[4], asthma[5], uterine fibroids[6] (UFs), and obstructive gastrointestinal diseases[7]. In normal tissue, SMCs are typically organized in a multi-layered structure with highly aligned cells whose architecture is tailored for optimal contractile function. Abnormalities in this organization are associated with a number of disorders. A prominent example is the case of UFs, common benign smooth muscle tumors that afflict up to 60–80% of all women[8], where SMCs organize into three-dimensional spheroid-like structures. The factors driving the changes in SMC organization during UF formation remain unclear.

A number of studies over the past few decades have reported that SMCs cultured on flat surfaces can under certain conditions spontaneously form three-dimensional structures that have alternately been referred to as hills, mounds, or hill-and-valley (HV) patterns[9–11]. These structures have been reported in SMCs from different species including human[12], pig[13], rat[14–16], and cow[17]. In some cases, the HV patterns transition to spherical "ball-like" clusters[18] that can either remain adherent to the substrate from which they emerged or become suspended in the culture medium. Interestingly, ball-like structures that originate from SMCs derived from human UFs revert back to HV patterns upon treatment with gonadotrophin releasing hormone (GnRH) in culture[19], suggesting that these three-dimensional structures may constitute a useful in vitro model for fibroids. Remarkably, despite the repeated and widespread observations, the physical mechanisms governing the spontaneous emergence of these intriguing three-dimensional SMC clusters and the dynamics of their evolution remain unexplained.

Here we use a combination of in vitro experiments and physical modeling to elucidate the mechanisms governing the spontaneous emergence and dynamics of three-dimensional SMC clusters from flat two-dimensional SMC layers. We show that the clusters initiate when SMC contractile forces overcome adhesion forces to the substrate, which only occurs once a critical SMC density has been attained. We also show that cluster formation occurs through a precise sequence of physical events whose dynamics we characterize both experimentally and using models that rely upon the well-established analogy between the spreading of cellular aggregates and the dewetting phenomena of liquid droplets[20–22].

## Results

Culturing SMCs on a fibronectin-coated flat glass surface leads to the spontaneous emergence of three-dimensional clusters once a sufficiently high cell density is attained (Supplementary Movie 1). The cluster diameter varies over a wide range of ~100 μm to ~3 mm. As shown in Fig. 1a and schematized in Fig. 1c, SMCs within clusters exhibit two distinct patterns of organization. Along the peripheral region of the cluster, particularly in the bottom portion[21], cells are oriented radially, have elongated nuclei, and express abundant F-actin, smooth muscle actin (SMA), and myosin heavy chain (MHC), consistent with radial force generation associated with the anchoring to the surrounding SMC layer (Supplementary Fig. S1a). On the other hand, SMCs in the inner region of the cluster assume an architecture that resembles a bird nest with nuclei oriented circumferentially and strikingly lower levels of F-actin, SMA and MHC expression as well as low expression levels of the extracellular matrix (ECM)

proteins collagen IV and laminin (Supplementary Fig. S1a–c), suggesting minimal force exertion and cell-substrate interaction. Interestingly, the centermost portion of the cluster often contains a zone of lower cell density (Fig. 1a, b, Supplementary Fig. S1d) that spans a portion of the height of the cluster. Live/dead staining revealed that the vast majority of the cells were viable (Supplementary Fig. S2).

Time-lapse imaging revealed that cluster formation occurs through the following sequence of specific physical events (Fig. 1d): (1) contractility-induced formation of a breach or hole in the flat SMC layer (Supplementary Movie 2), presumably due to an imbalance in the exerted contractile forces as a result of a locally elevated SMC concentration; (2) cellular retraction after hole formation, leading to a bumpy SMC "aggregate" that nevertheless remains fully attached to the substrate over its entire contour (Supplementary Movie 3); (3) additional contraction of the aggregate until a portion of it tears off from the substrate, giving rise to a well-defined three-dimensional cluster (Supplementary Movie 4); (4) rounding-up of the typically oblong cluster into a hemispherical shape (Supplementary Movie 5); and (5) cluster stabilization in its final hemispherical shape until the end of the observation (Supplementary Movie 6). Two clusters that come in contact are sometimes observed to fuse into a single cluster (Supplementary Movie 7). The physics of the different steps outlined above are described next.

### Hole development in the SMC layer and aggregate formation.

We investigated cluster formation at three different SMC seeding densities: low, medium, and high corresponding respectively to 40,000, 80,000, and 160,000 cells/cm$^2$ (see Methods). For the high-density group, cluster formation initiates within 24 h of seeding with the spontaneous development of holes in the SMC layer (Fig. 2a). In the early stages, the hole may still be partially covered with cells under high strain due to contractile forces from surrounding cells (Fig. 2a).

We quantified the velocity field in the cell sheet surrounding the hole using particle image velocimetry (PIV) (Fig. 2b, c). The hole widens as cells pull in opposite directions with a progressively increasing velocity. When the cell velocity on either side of the hole attains a peak (typically ~150 μm/h), the cells extending over the hole tear off, which has the effect of releasing the tension and stabilizing the hole width. This is accompanied with an overall reduction in cell velocity below ~20 μm/h.

We also characterized the evolution of the size and shape of the hole (Fig. 2d). This evolution can be modeled as the brittle fracture of a viscoelastic material where the dynamics are dominated by elasticity, similar to the work of Tabuteau et al.[23]. The profile of the hole is described by the so-called "De Gennes viscoelastic trumpet"[24,25], a general result for the fracture of viscoelastic materials. In the region closer to the fracture advancement edge, the profile is a parabola of equation $y = A\sqrt{x}$, where the symmetry axis $y$ of the parabola is aligned with the direction of hole propagation and the constant $A \sim \sqrt{\frac{G}{E}}$ depends on the adhesion energy $G \sim 10^{-2}$ J/m$^2$[23] and the elastic modulus of the SMC cell sheet $E \sim 2$ kPa. The latter estimate is based on the reported elastic modulus of single SMCs of 10 kPa[26] and of experiments with epithelial cells that indicate that the elasticity of a cell sheet is about 5 times smaller than that of a single cell[27]. Modeling the cell sheet as an elastic sheet, the opening dynamics of the hole width $a$ are driven by elastic recoil and resisted by friction with the substrate. Cell elastic recoil scales as $Eh(R_c - a)$, where $h$ is the SMC layer thickness and $R_c$ is the cell size. Friction scales as $kR_c^2 \frac{da}{dt}$, where $k$ is the cell-substrate friction coefficient. Balance between these two forces yields a differential equation

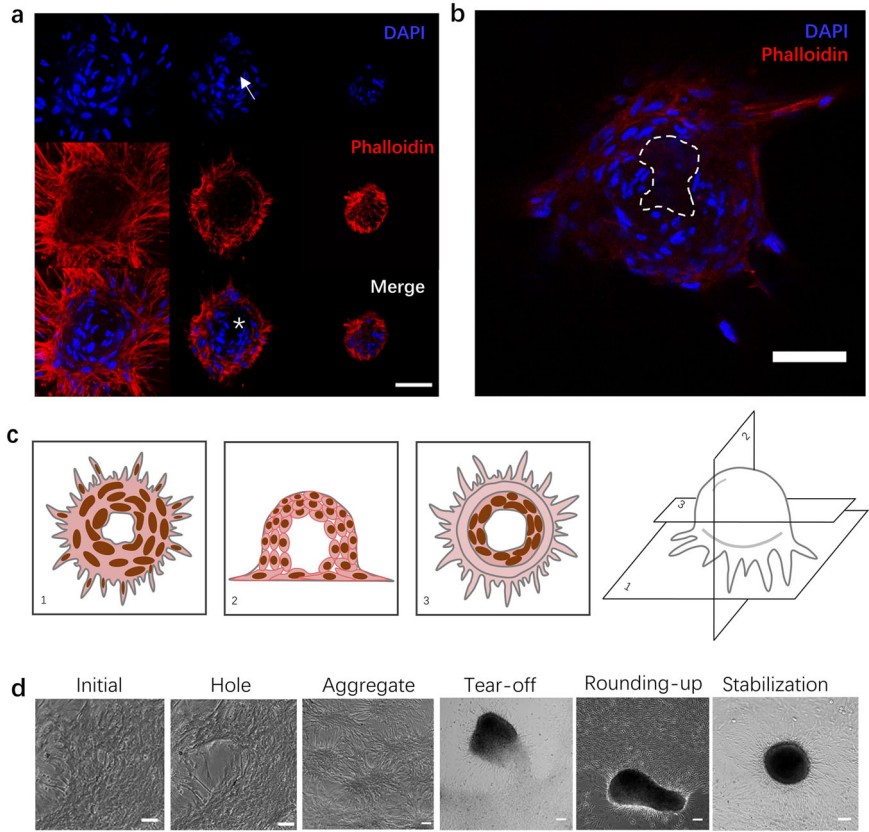

**Fig. 1 Cellular organization in SMC clusters and sequence of steps leading to cluster formation. a** Confocal images of cellular organization within a cluster. From left to right: z-stack from bottom to top of a cluster for the nucleus (DAPI channel; top row), F-actin (phalloidin channel; middle row), and both channels combined (bottom row). Note the lower cell density zone in the center of the cluster (white arrow on DAPI image; white * on merge image). Scale bar: 50 μm. **b** A confocal optical section through the midplane of a cluster to more clearly illustrate the lower cell density zone in the center (dashed contour). Scale bar: 50 μm. **c** Schematic of cellular organization within SMC clusters. **d** Phase contrast images of the sequence of five distinct events leading to cluster formation. From left to right: initial condition, crack development, aggregate formation, aggregate tear-off, cluster rounding-up from oblong to circular cross-section, and cluster stabilization. Scale bars: 100 μm.

whose solution is:

$$a = a_{\max}\left[1 - \exp(-Ct)\right] \quad (1)$$

where $C \sim \frac{Eh}{kR_c^2}$ is a constant. By fitting the experimental data (Fig. 2d), we obtain experimentally $a_{\max} \sim 200\,\mu m \sim R_c$ and $C \sim 2 \times 10^{-3}\,s^{-1}$, which enables us to estimate a cell–substrate friction coefficient of $k \approx 10^9$ Pa s m$^{-1}$, similar to the friction coefficient value reported previously[28,29]. Along its length, the hole opens in successive phases, corresponding approximately to one cell length[30] (Fig. 2d). According to our theoretical interpretation, the time to advance one phase should then scale as the stretching time $1/C \sim 500$ s, which is consistent with the observations (5–10 min). This hole development phenomenon is strongly reminiscent of the dewetting of a cell monolayer described by Douezan et al[28,31,32]., although here the dominant driving force is cell elastic recoil, rather than intercellular adhesion.

The nucleation sites of the holes appear to be related to local fluctuations in cell density (Supplementary Movie 8). These fluctuations, however, need not be very large as we have observed holes forming both in zones where cell density appears to visually be quite uniform as well as in regions where local density variations are more obvious. An established hole represents a valley in the commonly described HV SMC pattern. Once the hole expands, it ultimately forms a ring, leading to a topological transition with the formation of an "aggregate", a multilayer hill-like structure with a dense central portion and radially outward-pointing cells at the periphery (Supplementary Movie 3 and Supplementary Fig. S3a). These aggregates are highly dynamic as a result of gradients in cell pulling forces. PIV analysis revealed considerably higher cellular velocities in the monolayer (~30 μm/h) opposite to the direction of aggregate movement (Supplementary Fig. S3b, c).

**Cluster rounding and stabilization.** Driven by cell contraction, aggregates subsequently tear off from the substrate and form independent clusters (Supplementary Fig. S4). These clusters are mobile on the substrate as they progressively round up (Fig. 3a). We analyzed the evolution of cluster size and shape during this rounding-up process as well as cluster motion (Supplementary Fig. S5 and Supplementary Note 1) by imaging the cluster every 20 min (Supplementary Note 2). Cluster morphometric data were obtained through automatic detection of the bright contour in the images (Fig. 3a). Over 24 h, the cluster projected area decreased by about 40% (Fig. 3b) and its circularity (defined as $4\pi A/P^2$, where A is the cluster area and P is its perimeter), increased from 0.75 to almost 1 (Fig. 3c), thus indicating that the cluster becomes more compact and round over time. Similar to previous models[33], we interpret the dynamics of rounding up as a phenomenon driven by cluster surface tension, arising from both cell contractility and adhesion, and resisted by cluster viscous dissipation.

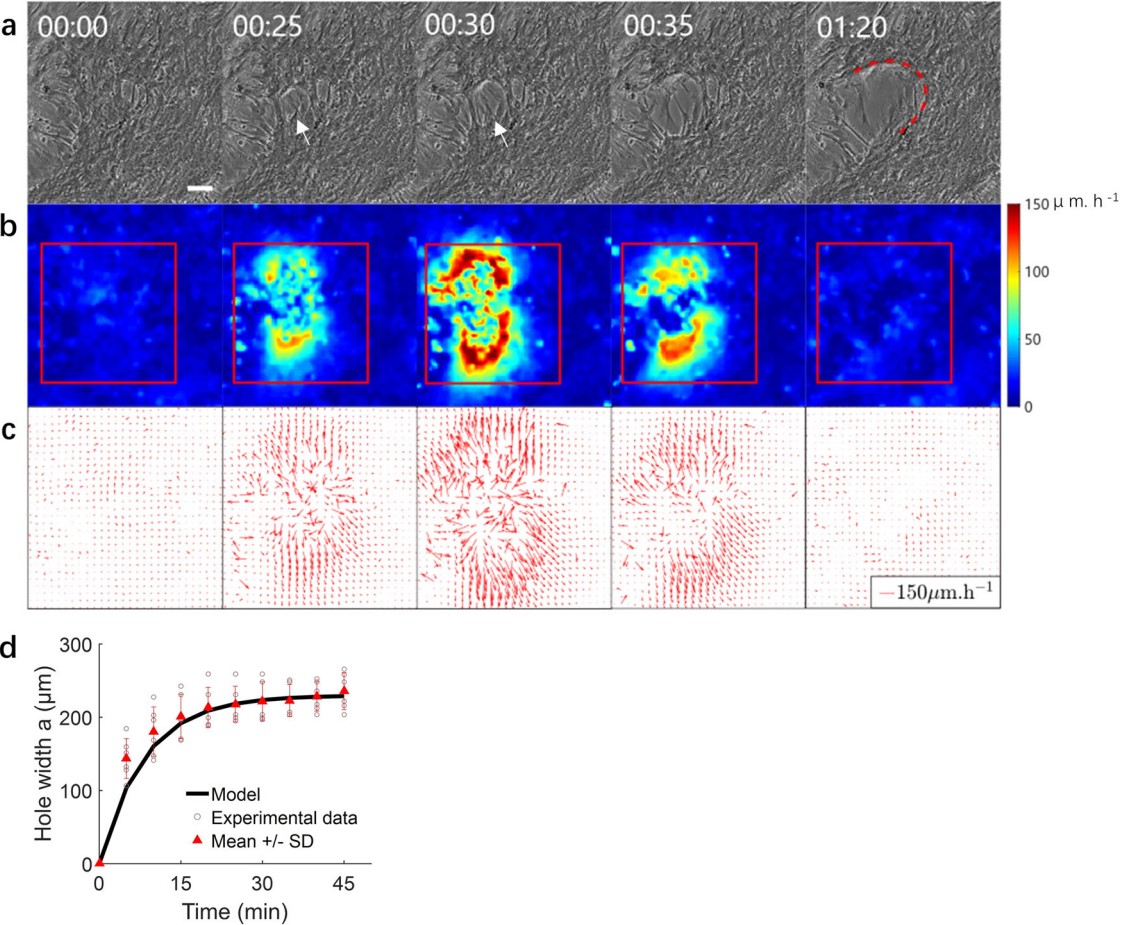

**Fig. 2 Dynamics of hole development. a** Phase contrast images of SMCs at different times pre and post-hole development. White arrows denote the position of hole initiation. Fit of the experimental shape of the hole edge by a parabola (red curve) whose equation is indicated in the text. Times are in hours and minutes. Scale bar: 100 μm. **b** Heat map of SMC tissue speed corresponding to the velocity field shown in (**c**). **c** Velocity vector field in the SMC tissue showing the larger velocities in the hole vicinity. **d** Time evolution of hole width. Comparison between the mathematical model of hole width evolution (solid black curve) to experimental data (brown circles), mean ± SD in red, $N = 6$. The stretching time ($1/C$) of hole opening can be observed at the beginning of opening. Interpretation of the constant $C$ is indicated in the text.

Mathematically,

$$\gamma \dot{S} = -\eta \int_V \overrightarrow{\nabla} \vec{u} : \overrightarrow{\nabla} \vec{u} \, dV \qquad (2)$$

where $\gamma$ is the surface tension coefficient, $\eta$ the cluster viscosity, $S$ and $V$ the cluster surface area and volume, and $\vec{u}$ is the velocity field within the cluster. We assume the cluster shape to be a prolate spheroid of major axis $a$ and minor axis $c$. By considering the spheroid to be sufficiently close to a sphere, we can obtain an expression for the velocity field and obtain a closed-form approximate solution to Eq. (2) (Supplementary Note 3):

$$In\left(\frac{\epsilon}{\epsilon_0}\right) = -\frac{15}{56} * \frac{\gamma}{\eta} * \frac{t}{r} \qquad (3)$$

where we have defined $\epsilon = \frac{(c-a)}{c+a}$ and $r = \frac{2a+c}{3}$.

A fit of Eq. (3) to the experimental data is shown in Fig. 3d, yielding $\frac{\gamma}{\eta} \approx 10^{-8}$ m/s, comparable to that reported for the rounding up of cellular aggregates of other cell types[31]. Once clusters round up, they stabilize in size and shape (Supplementary Fig. S6, Supplementary Note 4) unless they fuse with other clusters.

**Cluster fusion**. The interaction of two clusters that are sufficiently close to one another often leads to fusion (Fig. 4a). The

fused cluster subsequently goes through the "rounding-up" and "stabilization" steps described above for single clusters.

Cellular spheroid coalescence has been described by analogy with the fusion of two viscous liquid drops[33,34]. Recent developments have investigated the dynamics of spheroid fusion and how these dynamics are affected by cell proliferation[35] as well as by cell migration and the presence of ECM[36]. To analyze the fusion process, we quantify the evolution of the "neck" radius $\rho$, i.e., the radius of the contact region between two adjacent clusters as seen in projection on the experimental images. The observed neck evolution is reasonably fitted by the classic viscous sintering theory of Frenkel[37], which has been widely used to describe cell aggregate fusion:[33,38–40]

$$\frac{\rho^2}{R_0} = \frac{\gamma}{\eta} * t \qquad (4)$$

Here, $R_0 = \mathrm{sqrt}\left(\frac{A_1 + A_2}{2 * \pi}\right)$, with $A_1$ and $A_2$ the initial cluster areas, $\gamma/\eta$ is the ratio of surface tension to viscosity, and t is the elapsed time since the beginning of the fusion process (Fig. 4b, c). We note that even if the cluster behaves as a viscoelastic material, the fusion dynamics at times longer than the viscoelastic time (minutes to hours) remain appropriately described by the viscous sintering model[41,42]. By fitting Eq. (4) to the experimental observations of aggregate fusion, we obtain a value of $\gamma/\eta$ of the

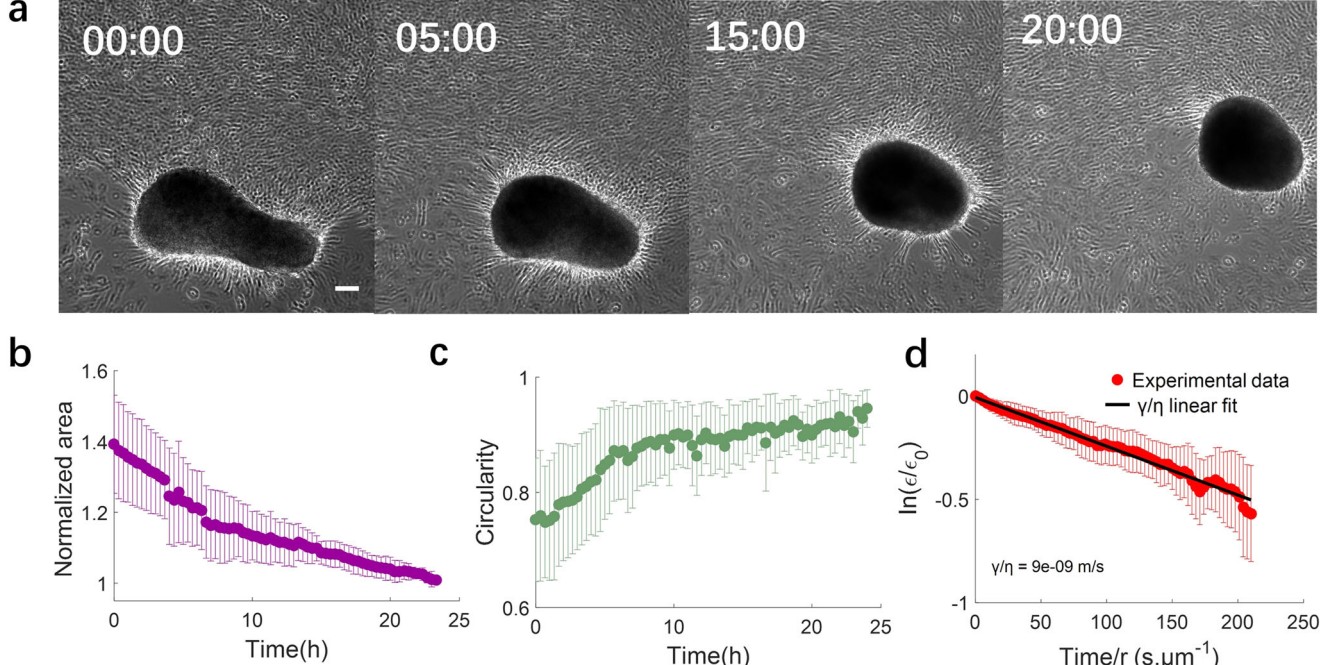

**Fig. 3 Cluster rounding dynamics. a** Phase contrast images of a SMC cluster at different times showing progressive rounding. Times are in hours. Scale bar: 100 μm. **b** Time evolution of normalized cluster projected area. **c** Time evolution of cluster circularity defined as $4\pi A/P^2$, where $A$ is cluster area and $P$ is cluster perimeter. **d** Normalized cluster/spheroid anisotropy, allowing determination of the value of the ratio of surface tension to viscosity ($\gamma/\eta$). $\gamma/\eta$ ~$10^{-8}$m/s. Equation is indicated in the text. **b–d** Mean ± SD, $N = 7$.

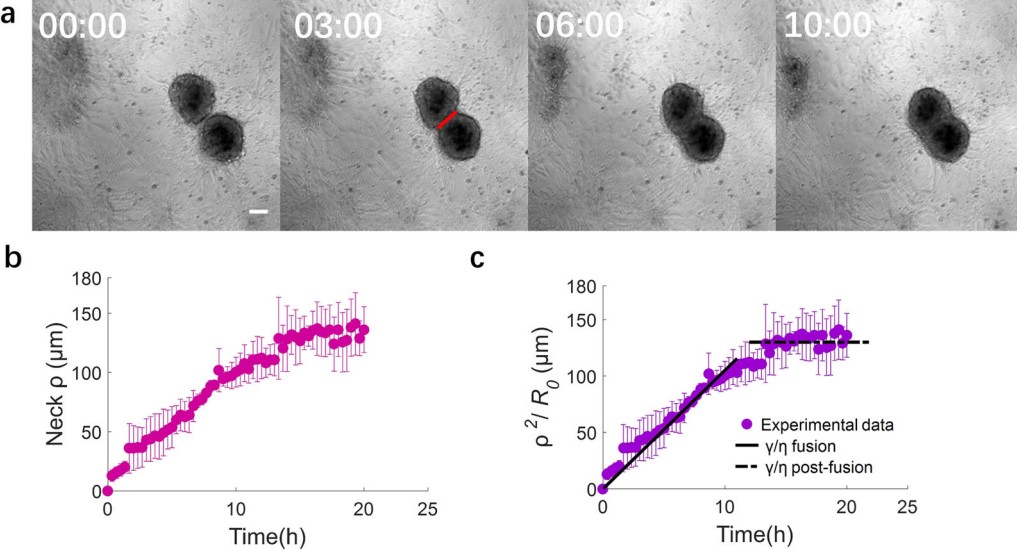

**Fig. 4 Cluster fusion. a** Phase contrast images of two fusing SMC clusters at different times. Times are in hours. Red line: $2\rho$, where $\rho$ is the "neck" radius. Scale bar: 100 μm. **b** Time evolution of "neck" radius $\rho$ for the fusion of pairs of clusters having approximately the same size (diameter ~200 μm). **c** Time evolution of the normalized neck radius. During fusion, $\gamma/\eta = 2.9 \times 10^{-9}$m/s. The "neck" radius stabilizes at the value of the cluster radius (~130 μm). Equation for $\gamma/\eta$ is indicated in the text. Panels b and c: mean ± SD, $N = 3$).

order of $2.9 \times 10^{-9}$ m/s. Taking into account possible scaling factors that arise from our simplified geometrical treatment of the problem, this value of $\gamma/\eta$ appears consistent with that obtained for the rounding-up process. Our findings thus suggest that both the "rounding-up" and "fusion" phenomena are governed by the same physics of an effective surface tension driving the morphological change, whose speed of evolution is limited by viscous dissipation. Interestingly, the apparent capillary velocity $\gamma/\eta$ for fusion is one order of magnitude smaller than that of rounding-up. We attribute this difference to the presence of ECM

especially at the cluster periphery, which slows down the fusion process (Supplementary Fig. S1b, c). This observation is consistent with previous reports of a role for the ECM in cell spheroid fusion dynamics[36].

After stabilization of the fused cluster, we observe a post-fusion rounding-up process (Supplementary Movie 9). We quantify the post-fusion rounding-up in the same way as done for the rounding phase (Supplementary Fig. S7). We quantify the $\gamma/\eta$, the ratio of surface tension to the viscous dissipation, as well as the normalized projected area and the circularity for the fused

clusters to analyze the post-fusion rounding-up dynamics. The post-fusion cluster rounded up within 16 h (Supplementary Fig. S7a). The projected normalized area decreases by approximately 20% (Supplementary Fig. S7b), and the circularity increases from 0.85 to 1 (Supplementary Fig. S7c). A linear fit to the experimental data yields $\frac{\gamma}{\eta} \approx 10^{-8}$ m/s (Supplementary Fig. S7d), which is of the same order of magnitude as the ratio of the normal rounding-up cases. These results demonstrate that the initially ellipsoidal fused cluster exhibits a rounding-up speed that is similar to that of individual clusters.

**Prerequisite conditions for SMC cluster formation**. SMC cluster formation as detailed above requires two prerequisite conditions: a sufficiently high cell density and sufficient levels of SMC contractility. The role of each of these two conditions is characterized next.

SMC density determines both the dynamics of cluster formation and the pre-cluster organization of SMC sheets necessary for initiating a cluster. We investigated the dynamics of cluster formation for the three different cell seeding densities tested (40,000, 80,000, and 160,000 cells/cm²) (Supplementary Fig. S8). We divide the clusters into two types (Supplementary Fig. S9): "developed clusters", which have a clear and complete contour, and "developing clusters", which have a partially developed contour. Not surprisingly, cluster formation occurs most rapidly at the highest cell seeding density for both types of clusters (Fig. 5a, b and Supplementary Fig. S8b). Furthermore, cluster formation at the lower seeding densities occurs only after a delay, suggesting that a critical cell density needs to be attained *via* cell proliferation for cluster formation to initiate. For both types of clusters, the number of clusters increases for several days until it reaches a maximum and then decreases before ultimately stabilizing at a plateau. For developing clusters, the increase is due to cluster formation because of cell proliferation, whereas the decrease corresponds to developing clusters maturing into developed clusters. For developed clusters, the increase is driven by the maturing developing clusters, while the decrease is due to cluster fusion once cluster density becomes sufficiently high for neighboring clusters to interact. The time evolution of the total number of clusters (i.e. developing and developed) exhibits similar overall behavior (Fig. 5c) once a shift of 3 days has been introduced for the low and medium seeding densities, so that $t = 0$ for all experiments is the time when cluster formation starts. With this shift, all experimental curves are approximately similar. This master curve first rapidly increases to reach a maximum, and then decreases to finally stabilize at a plateau.

The master curve in Fig. 5c can be described by the following population dynamics differential equation:

$$\frac{dN}{dt} = p_1 - \alpha(t)p_2 N, \tag{5}$$

where $N$ is the total number of clusters at time $t$, $p_1$ is the rate of cluster formation, and $p_2$ the rate of (relative) cluster number reduction, mainly arising from cluster fusion. The time-dependent parameter $\alpha(t)$ is introduced to account for a reduced rate of cluster number at short time, due to the time required for clusters to meet each other and fuse. We assume a linear increase of the dissociation rate over time, $\alpha(t) = \min(t / t_s, 1)$, with $t_s$ being a certain stabilization time. The solution to this differential equation agrees closely with the experimentally observed shape. Figure 5c shows a comparison between the computational fit and the experimental data with fitted model parameter values of $p_1 = 50$ clusters/day, $p_2 = 1.5$ day$^{-1}$ and $t_s = 8$ days.

It would be instructive to characterize the organization of our SMC layers prior to cluster formation for the different cell densities. To this end, we quantify the extent of actin stress fiber alignment by computing the scalar order parameter Q for non-polar particles[43] (Supplementary Fig. S10):

$$Q = \sqrt{<\cos 2\theta>^2 + <\sin 2\theta>^2} \tag{6}$$

where $\theta$ is the angle between the stress fiber direction and a reference direction (the horizontal direction in this case). Q ranges from 0 for a completely disordered case to 1 for a perfect nematic order where all stress fibers are aligned in the same direction. At low cell density, SMCs exhibit a wide range of orientations (Figs. 6a, 6c i) as indicated by the small value of $Q = 0.18 \pm 0.07$ (Fig. 6b). At medium density, SMCs become confluent and coordinate with local alignment of stress fibers (Figs. 6a, 6c i), leading to considerably increased order ($Q = 0.51 \pm 0.16$ (Fig. 6b)). At high density, SMCs rapidly form multiple layers (Figs. 6a, 6c ii), typically three (Supplementary Fig. S10), prior to cluster formation, and actin stress fibers within each layer are highly aligned (Supplementary Fig. S10b), with an order parameter approximately equal to unity ($Q = 0.87 \pm 0.13$ (Fig. 6b)). These findings are consistent with previous studies[43–45].

Our results indicate that SMC cluster formation only initiates once a critical cell density is attained and the cells have formed several layers. However, cell density per se is not sufficient to initiate cluster formation. The additional required component is sufficiently elevated cell contractility in order to overcome cell-substrate adhesion as depicted schematically in Fig. 6c. Such force disequilibrium would lead to the tearing-off of the SMC cell sheet

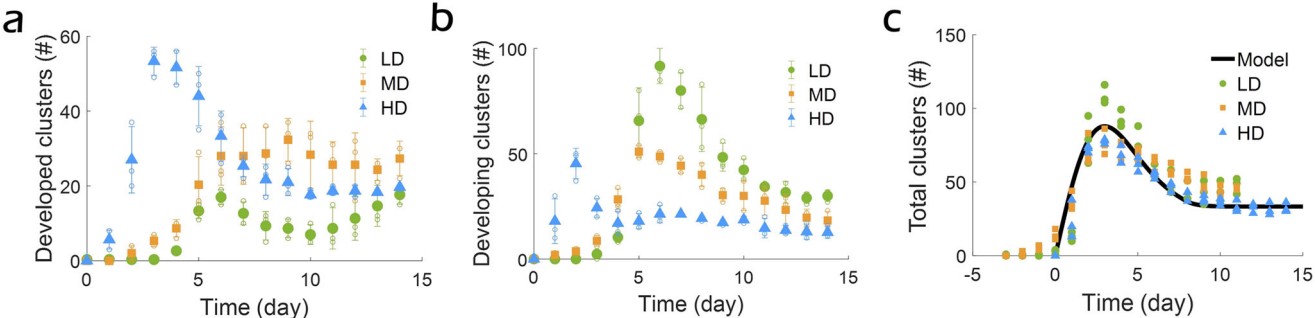

**Fig. 5 Dynamics of cluster evolution. a** Time evolution of the number of developed clusters for the different cell seeding densities. The maximum number of developed clusters is reached at days 14, 9 and 4 for the low (LD), medium (MD), and high (HD) seeding densities, respectively. **b** Time evolution of the number of developing clusters for the different cell seeding densities. The maximum number of developing clusters is reached at days 6, 5 and 2 for the low (LD), medium (MD), and high (HD) seeding densities, respectively. **a, b** Mean ± SEM, N = 3. **c** Time evolution of total number of clusters (sum of developed and developing clusters) and physical model fit. Three experiments for each density are shown.

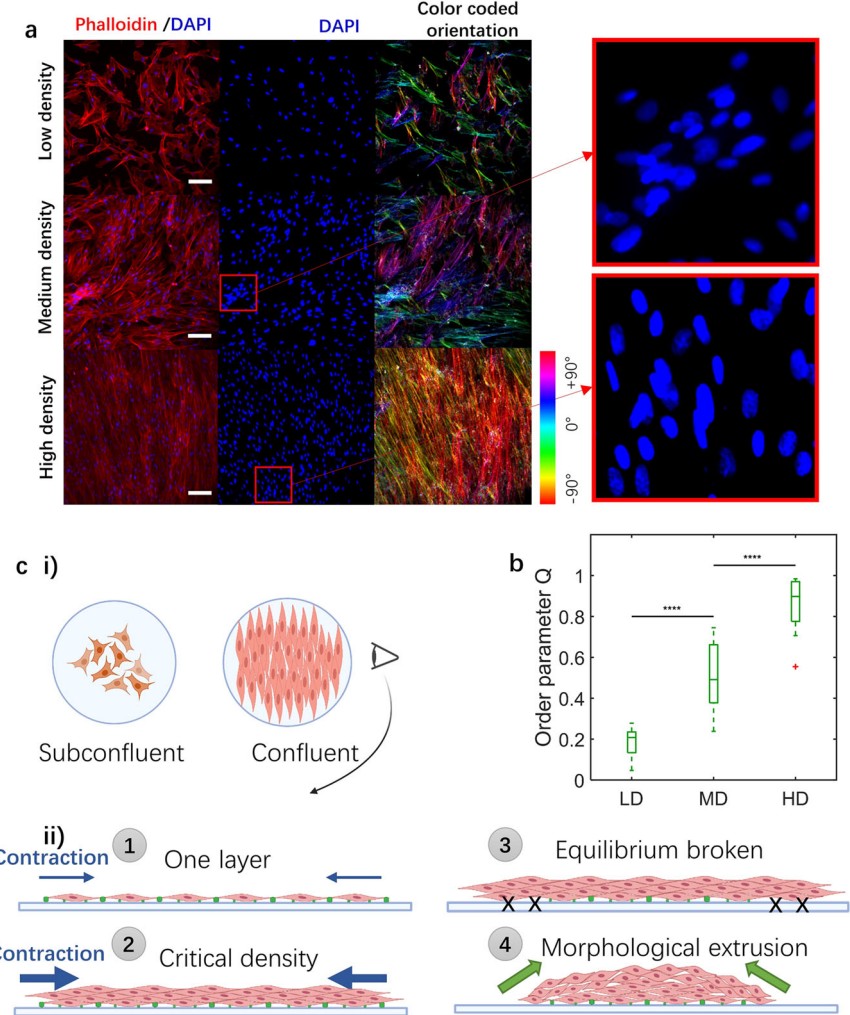

**Fig. 6 Pre-cluster SMC actin stress fiber organization. a** Immunostaining images of samples with different seeding densities 24 h post-seeding. DAPI (blue) shows SMC nuclei, while phalloidin (red) shows F-actin. Nuclear overlap in some regions (see insets) demonstrates that cells begin to form local multilayer zones at the medium density. Scale bars: 100 μm. **b** Order parameter values for the low (LD), medium (MD), and high density (HD) cases. *P* value is calculated using a one-way ANOVA, ****$p \leq 0.0001$, $N_{LD} = N_{MD} = N_{HD} = 15$. Error bars represent standard deviations. **c** Schematic of density- and contraction-induced cluster formation mechanism. (i) Subconfluent field represents low density samples. Confluent field represents medium density samples. (ii) Schematic side view of morphological evolution with different seeding density. 1, One confluent layer of SMCs, 2 three to four layers (critical density) of SMCs, 3 Contraction force overcomes cell-substrate adhesions. 4 Morphological extrusion due to cell contraction and subsequent detachment from the substrate. Blue arrows represent contraction force of SMC layer. Multilayers have higher contraction force. Green dots represent cell-substrate adhesion.

from the substrate. In support of this model, SMC contractility inhibition using blebbistatin (5 μM) significantly slows down cell detachment from the substrate (~30% decrease in detached area at day 3) and delays the formation of clusters (Supplementary Fig. S11a), A higher dose of blebbistatin (10 μM) completely inhibits cluster formation at day 2, and at day 3, the cell detached area is more than 20 times smaller than in untreated controls, confirming the important role that SMC contractility plays in cluster formation. We note that we attempted to probe the effects of even higher doses of blebbistatin (15 and 20 μM) on cluster formation; however, those doses led to detachment of the entire cell sheet, presumably due to the fact that high blebbistatin concentrations compromise the strength of cell adhesion to the substrate.

As an additional test, we investigated the dynamics of SMC cluster formation when the cells were cultured on very soft gels (Young's modulus of ~2 kPa) on which their ability to generate large contractile forces is greatly diminished. The results demonstrated that when plated at the same seeding density, cluster formation on the soft gel required 3 days longer than on plastic (Young's modulus ~2.7 GPa). Applying the model described in the "Rounding-up" section for the dynamics of cluster formation yields a value of γ/η that is 3 to 4 times smaller for the soft gel than for plastic (Supplementary Fig. S11b), consistent with a slower rate of cluster formation. These findings are consistent with the central role for contractility as the driving force for SMC cluster formation.

## Discussion

The spontaneous formation of three-dimensional SMC clusters, referred to elsewhere as HV patterns and ball-like structures, from flat cellular layers has been described qualitatively in many studies over the past few decades; however, the physical processes giving rise to these clusters have not been described. The present study provides the first detailed quantitative characterization and

physical modeling of the events leading to the formation of SMC clusters.

Our experiments indicate that cluster formation occurs through a sequence of physical events whose dynamics and underlying mechanisms we characterize by conceptualizing SMC clusters and their precursor structures (referred to as aggregates) as active liquid droplets. This physical formulation allows us to explain the observed dynamics of: (i) hole development in the flat SMC layer, analogous to brittle viscoelastic fracture followed by dewetting, (ii) cluster shape evolution, analogous to droplet rounding-up, and (iii) cluster fusion, analogous to droplet coalescence. Liquid droplet models have indeed been widely used to describe biological self-assembly at different scales, from actin bundle assembly[46], to single mitotic cell rounding[47], and aggregate development of different cell types[48]. The model's predictions are in agreement with observations and provide estimates of the material properties of SMC clusters. At a population scale, we describe the evolution of the number of clusters by a population dynamics model that accounts for cluster formation and cluster fusion.

The speed $\gamma/\eta$ of cluster fusion is an order of magnitude smaller than that of cluster rounding. We propose that this is attributable to the dense ECM distribution on the periphery of the clusters (Supplementary Fig. 1b, c). The fusion process involves the interaction and remodeling of the ECM within the contact area between the two clusters, which increases the viscosity of the system. Thus, it appears reasonable that the fusion speed would be slower than that of the rounding of an individual cluster.

A recent study of stem cell aggregates reported the observation of arrested coalescence during aggregate fusion[48]. In that case, fused aggregates maintained an ellipsoidal shape for several hours and did not rapidly round up to a spheroid. In the present study, the fused clusters exhibit liquid-like properties that lead to rounding with dynamics that are of the same order of magnitude as those of the rounding of individual clusters. The difference between the "arrested coalescence" seen in stem cell aggregates and our SMC cluster "liquid-like rounding" may be attributable to differences in internal structure[49].

Clusters in SMC cultures have been reported in the literature on multiple occasions[10,18,26] and appear to have anecdotally been considered as a sign of "high passage" cells that have exceeded their "usefulness" for standard SMC in vitro experiments. Consequently, the processes of formation and evolution of these clusters have received limited attention thus far. The current study demonstrates that these clusters can form at relatively low passage (3 to 10 used here) as long as a sufficiently high cell density is attained. It should also be noted that cell contractility is an important consideration in cluster formation. For instance, in limited experiments, we were able to observe that clusters fail to form when using human umbilical vein SMCs which are presumably less contractile than the aortic SMCs used in the present study. The fact that blebbistatin treatment significantly slows down cluster formation provides further support for the importance of contractility in this process.

The potential biological and physiological relevance of SMC clusters remains to be established. As already described, cells in SMC clusters are oriented radially along the periphery of the cluster and circumferentially in the interior region with a near "cell-free" zone in the center. This organization shares some similarities with that of UFs that have been reported to exhibit regions of radial and circumferential cell orientation[50]. Moreover, the enrichment of SMA, MHC, and ECM proteins at the periphery of SMC clusters bears some resemblance to the fibrous caps of atherosclerotic paques[51–53], while the lower cell density zone in the center of the clusters is somewhat akin to the "necrotic cores" in both atherosclerotic lesions[54,55] and UFs[56]. It would therefore be particularly interesting to probe if SMC clusters may serve as a useful in vitro model for studying aspects of atherosclerotic plaque and UF development and density-driven SMC hypertrophy and hyperplasia.

## Methods

**Cell culture**. Bovine aortic smooth muscle cells (SMCs; Cell Applications, Inc., San Diego, CA, USA) in passages 3-10 were cultured using standard procedures in bovine SMC growth medium (Cell Applications) and maintained at 37 °C and 95% air/5% $CO_2$.

**SMC cluster formation and visualization**. Cells were seeded on either tissue culture-treated well plates (Costar, Corning Inc., New York, NY, USA) or fibronectin-coated glass surfaces at either "low", "medium", or "high" density, corresponding respectively to 40,000, 80,000, or 160,000 cells/cm$^2$. The low density is considered the baseline seeding density and yielded a confluent SMC monolayer after ~24 h. The medium density corresponds to a confluent monolayer upon cell adhesion, whereas the high density results immediately in 2 to 3 overlapping cell layers. Except for the different initial densities, all experiments were performed under the same conditions. Cellular evolution was monitored over 14 days.

Bright-field images were acquired using an inverted microscope (Nikon ECLIPSE Ti2, Tokyo, Japan) equipped with a 10X objective. The acquisition rate was 1 frame every 20 min for most observations and 1 frame every 5 min for the "hole development" experiments whose dynamics were faster. Images (2048 × 2044 pixels) were exported in TIFF format and analyzed using FIJI.

**Preparation of coated glass substrates**. Small (7 mm-diameter) holes were drilled into a 1 cm-thick polydimethylsiloxane (PDMS; Sylgard-to-catalyst ratio of 10:1) sheet to accommodate thin glass coverslips. The PDMS and coverslips were sterilized in ethanol for 5 min, dried at room temperature, plasma treated for 45 s, and then assembled together by pressure application. The assembly was then sterilized in ethanol for 5 min and dried at room temperature. The 7 mm wells so created were coated with a 0.1 mg.mL$^{-1}$ solution of GIBCO$^{TM}$ Fibronectin Bovine Protein (Thermo Fisher Scientific, Boston, MA, USA) in PBS (pH 7.4) for 45 min at room temperature. The coated surfaces were then rinsed with PBS (pH 7.4) and used within 2 h.

**Preparation of coated gel substrates**. Soft gel substrates were prepared using the DOWSIL$^{TM}$ CY 52-276A&B gel formulation (Dow Corning, ratio 1:1) according to the manufacturer's specifications, leading to a gel with an elastic modulus of 1-3 kPa. The gel was poured into 48-well plates and maintained at 75 °C for 10 h. The gel surfaces were sterilized in ethanol for 5 min and dried at room temperature before coating with a 0.1 mg.mL$^{-1}$ solution of GIBCO$^{TM}$ Fibronectin Bovine Protein (Thermo Fisher Scientific) in PBS (pH 7.4) for 45 min at room temperature. The coated surfaces were then rinsed with PBS (pH 7.4) and used within 2 h.

**SMC cluster internal structure imaging**. The 3D organization of SMC clusters was imaged using a confocal microscope (Leica SP8, Wetzlar, Germany) equipped with a 63X objective. The cells were stained with DAPI and phalloidin to visualize nuclei and F-actin. Images were exported in TIFF format and analyzed using FIJI.

**Image analysis**. Cluster outlines were defined by their bright contours in phase contrast images. Cluster contours were tracked using the Analyze Particles FIJI plug-in. Cluster morphometric parameters including area, perimeter, centroid position, major axis length, and minor axis length were quantified using a custom-made MATLAB code.

**Blebbistatin treatment**. Blebbistatin was diluted in cell culture medium, and two concentrations were used: 5 μM and 10 μM. Blebbistatin was administered for one hour per day over three days (at the same time each day) after which samples were washed carefully with PBS 5 times and then cultured in cell culture medium.

**PIV analysis**. Particle image velocimetry (PIV) was used to obtain the velocity fields of SMC tissue. The PIV analysis was performed using the Matpiv package in MATLAB (https://www.mn.uio.no/math/english/people/aca/jks/matpiv/). A three-pass computation using a final window of 32 × 32 pixels (20.8 × 20.8 μm) was used. The overlap was set at 0.5. Aberrant vectors were detected and removed from the analysis when their magnitude exceeded the local mean value by three times the standard deviation (~5% of the data). The time interval between consecutive images was 5 min in the "hole development" experiments and 20 min in the other cases. For each frame, we computed the root-mean squared-velocity as

$V_{rms} = \sqrt{\langle V^2 \rangle}$, where $V$ is the magnitude of the velocity and the brackets denote an average over the entire field.

**Immunostaining**. Samples were fixed with 4% paraformaldehyde for 15 min and rinsed with PBS three times and then permeabilized with 0.1% triton solution in PBS (PBST) for 15 min. Non-specific binding was blocked using 3% BSA in PBST. Nuclei were stained with DAPI (SigmaAldrich, St. Louis, MO, USA; 1:20,000) and F-actin with Phalloidin594 (LifeTechnologies, Carlsbad, CA, USA; 1:200). The incubation time was 1 h.

**Statistics and reproducibility**. A minimum of three experiments were performed. All data are expressed as mean ± SD. Differences were assumed to be statistically significant for $p < 0.05$. Statistical comparison for the data on F-actin fiber orientation data was performed using a one-way ANOVA. Statistical comparison for the data on substrate rigidity data was performed using the Student t-test. All statistical analyses were conducted in MATLAB.

**Reporting summary**. Further information on research design is available in the Nature Portfolio Reporting Summary linked to this article.

# Data availability
The source data behind the graphs in the paper can be found in Supplementary Data. All other data that support the findings of this study are available from the corresponding author upon reasonable request.

# Code availability
The codes used in this study are available from the corresponding author upon reasonable request.

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

## Acknowledgements

The authors acknowledge all members of the Barakat group for their constructive input to the preparation of the manuscript. We thank Prof. Julien Husson for insightful discussions. This work was funded by an endowment in Cardiovascular Bioengineering from the AXA Research Fund (to A.I.B.) and a doctoral fellowship from Ecole Polytechnique (to X.W.).

## Author contributions

X.W. and A.I.B. conceived the study. X.W. conducted the experiments. D.G.R. developed the theoretical models. X.W. analyzed the experimental results with the help of D.G.R. X.W. developed the analysis tools. T.V. conducted the PIV analysis. X.W., D.G.R., and A.I.B. wrote the manuscript. X.W., D.G.R., T.V., P.S., and A.I.B. participated in reviewing and editing of the manuscript.

## Competing interests

The authors declare no competing interests.
