## [Peer Review File · Communications Biology]

Reviewers' comments:

Reviewer #1 (Remarks to the Author):

In their work entitled "Contractility-induced self-organization of smooth muscle cells: from multilayer cell sheets to dynamic three-dimensional clusters", Wang et al show and analyze the formation of 3D cell clusters of smooth muscle cells. The study is biologically relevant and interesting from the biophysical modeling perspective. However it is lacking in some key analyses and explanations detailed below, which the authors should address before it can be published.

1. The fracture model behind the hole opening, Eq. 1, is not detailed in the text, but instead the reader is referred to Ref. 22. To make the text self-contained, at least a brief explanation for the mechanical force balance that leads to Eq. 1 should be given here. (Fig. 1) It is also not clear why the profile of the hole is parabolic. Ref. 22 assumes a parabolic profile for drops falling under gravity, where the parabolic approximation seems more reasonable.
2. Do the authors have any thoughts on the nucleation sites of these holes? Are these random fluctuations in cell density, or can it be controlled experimentally? (Fig. 1)
3. How many clusters are considered for the rounding up analysis in Fig. 3? If the cluster rounding dynamics can indeed be described by a simple physical model balancing surface tension and viscous dissipation, the same timescale should emerge for different clusters when normalized by size, and a master curve (data collapse) should be seen for the plot in Fig. 3d. In other words, the surface tension and viscosity as material parameters should be similar for different clusters.
4. The same comment as the above applies to the cluster fusion data in Fig. 4. Is it possible to get more statistics than just three events?
5. The fused cluster should also relax to a round shape because of surface tension. Do the authors observe this? If not, why not? And if yes, then can they quantify this rounding up in the same way as done for Fig. 3?
6. The authors should quantify cluster size distribution, and rate of growth of mean cluster size with time, and see if this leads to a power law expected for surface tension driven growth.
7. The nematic order data in Fig. 6 is along expected lines, and doesn't have any particularly new message. Does the nematic orientation axis influence hole formation in any way? Have the authors looked for topological defects in the nematic order? These have been found to influence cell mounding or extrusion in layers of motile cells and could be interesting if found relevant in adherent cells as well.
8. Can the authors explain the radial organization of cells near the periphery of the cluster and circumferential along the central hole? Is this because of the differences in adhesion/anchoring that the cells have with the substrate vs a free interface?
9. The biological scope of the discussion seems a little narrow, given the emphasis on physical modeling. Similar dewetting/rounding up processes can also occur for single cells (such as cells just before dividing) and hole formation or closure could be related to wound healing or morphogenetic processes. Could the physical mechanisms identified by the authors apply to those situations, which also utilize cellular contractility and adhesion, even if the cell type is different? A discussion of this point could be interesting.

Minor:

1. In the panel for Fig. 2, the subfigures are not marked "a", "b", "c"... which makes reading the captions difficult. Also, I looked but couldn't find the "white arrow".
2. A similar analysis of cytoskeletal filament clusters as liquid droplets which relax to characteristic shapes and fuse/coalescence was performed in Weirich et al. PNAS 2017 which the authors may find relevant. (<https://www.pnas.org/content/early/2017/02/09/1616133114>)

Reviewer #2 (Remarks to the Author):

The manuscript by Wang et al provides a description of the formation of 3D clusters of SMs from a monolayer. This experimental work is supported by a semi-quantitative theoretical analysis. The initiation of the 2D to 3D transition is described as a brittle fracture and the growth of the 3D cluster as a dewetting process. This work is elegant by its simplicity. The authors only used timelapse video-microscopy, and they performed PIV and a few stainings on fixed samples for characterization. Still, they could propose an overall understanding of the process.

By itself, the analogy between 3D cell aggregate formation from a cell sheet and the process of liquid dewetting has already been reported several times. Unfortunately, while the authors keep on referring to papers from previous works from one author (DGR) and other colleagues, they even do not refer to the proper paper (line 162, the dewetting paper from the Brochard-Wyart team to cite is not 28 or 29, but Douezan, S., Brochard-Wyart, F. Dewetting of cellular monolayers. *Eur. Phys. J. E* 35, 34 (2012). <https://doi.org/10.1140/epje/i2012-12034-9>). The tendency to self-citation by the authors is highly regrettable. I would recommend the authors to do a proper bibliographic search and give credit to other groups. As an example, they could cite the work from the Trepas team (Pérez-González, C., Alert, R., Blanch-Mercader, C. et al. Active wetting of epithelial tissues. *Nature Phys* 15, 79–88 (2019). <https://doi.org/10.1038/s41567-018-0279-5>).

In the end, my two major concerns with this manuscript are the following ones:

1/ The paper lacks statistical analysis. One has the feeling that this phenomenon has been seen by chance and/or could be recorded properly only a couple of times. From that, the authors adapted an existing soft matter approach.

Moreover, even though the authors propose that hole opening is controlled by elastic recoil and friction with the substrate, the estimate of the parameters is very rough. For instance, it seems that they take the Young's modulus of individual cells instead of the one of cell sheet (that could be measured).

Founding the validity of the model on a derivation of the friction coefficient, which is more than one order of magnitude larger than previous reports. In other words, I think that, due to a lack of statistics and critical tests, the theoretical understanding of the process lies on a soft ground. In particular, I cannot figure out while contractility impairment (using blebbistatin or soft substrate) only introduces a lag in the process instead of inhibiting it.

Cluster rounding and coalescence of clusters should even not show up here, since these phenomena are known for ages. In particular, the droplet model for spheroid coalescence used by the authors is quite debated (see eg. Kosheleva, N.V., Efremov, Y.M., Shavkuta, B.S. et al. Cell spheroid fusion: beyond liquid drops model. *Sci Rep* 10, 12614 (2020). <https://doi.org/10.1038/s41598-020-69540-8>). It could have been of interest to test a more elaborate model by considering the aggregates as a viscoelastic material instead of a viscous liquid, as recently proposed (Dechristé G, Fehrenbach J, Grisetti E, Lobjois V, Poignard C. Viscoelastic modeling of the fusion of multicellular tumor spheroids in growth phase. *J Theor Biol.* 2018 Oct 7;454:102-109. doi: 10.1016/j.jtbi.2018.05.005).

2/ My second major criticism is that this work is sold as a work that unravels the underlying mechanisms of SMC-related diseases. This is the main focus in the abstract and in the introduction of the manuscript. To me, the present work is primarily a nice soft matter-inspired biophysical work that suffers from a lack of statistics, sensitivity discussion and thorough critical tests.

In conclusion, even though there is apparently nothing wrong in this paper, modeling is on a soft ground because several parameters are unknown and not directly measured, novelty is by far not obvious, and the biological motivation is a pretext of this "biological physics" work. I would therefore recommend publication in a more specialized soft matter journal

Reviewer #3 (Remarks to the Author):

Report on the Manuscript "Contractility-induced self-organization of smooth muscles cells: from

multilayer cell sheets to dynamic three-dimensional clusters"

X. Wang and coworkers report a study on the spontaneous 2D \rightarrow 3D transition of a layer of smooth muscle cells. They focus on the physics of this transition and quantitatively explain it, by invoking the sudden symmetry breaking (layer fracture) and the following dewetting.

In my opinion, this is a robust biophysical paper, which includes in vitro experiments, data analysis and modeling. However, I have noted some weaknesses in the article, which deserve to be reviewed.

Major points:

1. From the text, it is not clear how many times the experiments were reproduced. Figures 2, 3, 4 and 6 show single examples of data, and the corresponding fits. However, they do not provide a summary panel, reporting the fitting values obtained from different experiments and the distribution of such values.
2. Through the text, most of the results are given without errors (or standard deviations) and the number of samples. This information must be added.
3. A precise statement on reproducibility should be added. In particular, how many experiments have been done, how many succeeded, how many failed or gave different results (and why).
4. In movie #1 something weird happens between $t = 31h00'$ and $t = 31h20'$. It seems that two movies have been merged, as there are no common features between the two frames. In addition, but also brightness and contrast suddenly change.
I have hypothesized that the experimentalist changed some parameters after the fracture, such as the focus and the exposure time. Nevertheless, it is necessary to replace this movie with a better one, or – at least – provide a satisfying explanation of this sudden change.

Minor points:

1. The authors insist on the fact that a low-density area appears in the center of the clusters shown in fig. 1a. In the text, they should explain better why this feature looks so interesting.
2. Movie 6: it seems that after formation the aggregate keeps attracting more and more cells from the subjacent layer. Is it true?
3. Movie 7: the aggregates fuse, but do not fully merge. Do they ultimately form a unique aggregate, or a kind of interface remains between the two? If the fully merge, how long does it take? What is the mechanism of cell rearrangement, motility or death/division?
4. Figure 4c : check the unit: μm or μm^2 ?

Response to Reviewers

We would like to thank the reviewers for their insightful and constructive comments. We have endeavored to respond positively to each of their concerns, and we believe that the resulting changes have yielded a more complete and much improved manuscript. Below is a point-by-point response to the specific points raised by the referees. The reviewers' comments are shown in italics. All page numbers refer to the revised manuscript.

Reviewer #1

In their work entitled “Contractility-induced self-organization of smooth muscle cells: from multilayer cell sheets to dynamic three-dimensional clusters”, Wang et al show and analyze the formation of 3D cell clusters of smooth muscle cells. The study is biologically relevant and interesting from the biophysical modeling perspective. However it is lacking in some key analyses and explanations detailed below, which the authors should address before it can be published.

1. The fracture model behind the hole opening, Eq. 1, is not detailed in the text, but instead the reader is referred to Ref. 22. To make the text self-contained, at least a brief explanation for the mechanical force balance that leads to Eq. 1 should be given here. (Fig. 1) It is also not clear why the profile of the hole is parabolic. Ref. 22 assumes a parabolic profile for drops falling under gravity, where the parabolic approximation seems more reasonable.

As suggested by the reviewer, we have edited the manuscript to briefly explain how Eq. 1 is obtained (pp. 5-6). The equation arises from a balance between the cell elastic recoil force, which drives the opening, and viscous friction, which resists it. When detailing this expression in the manuscript, we also corrected the scaling of the elastic force, which should scale with the cell sheet thickness. This correction yields a revised prediction of the friction coefficient that is more consistent with literature values, as mentioned in the response to one of the points raised by Referee 2 below.

The parabolic profile is a result independent of our Eq. 1 (albeit consistent with it). The parabolic profile is a classical result in physics known as the "De Gennes viscoelastic trumpet". In a seminal article, De Gennes theoretically described the shape of a fracture opening in a viscoelastic material, which corresponds to a parabolic shape in the region closest to the location of fracture initiation. This is a general result valid for viscoelastic materials, whether a viscoelastic liquids as in Ref. 22 or a viscoelastic cell sheet as in our work. The parabolic profile is not due to gravity forces. This explanation is presented in the revised manuscript (p. 5).

2. Do the authors have any thoughts on the nucleation sites of these holes? Are these random fluctuations in cell density, or can it be controlled experimentally? (Fig. 1)

The reviewer raises an interesting point. In light of the fact that cell density appears to play a central role in the dynamics of hole initiation and cluster formation, we believe that the nucleation sites of the holes may indeed be related to local fluctuations in cell density. These fluctuations, however, need not be very large as we have observed holes forming both in zones where cell density appears to visually be quite uniform as well as in regions where local density variations are more obvious. At our current stage of understanding of the phenomenon, we are not able to control hole initiation experimentally. We simply observe its occurrence and characterize the dynamics of its evolution.

3. How many clusters are considered for the rounding up analysis in Fig. 3? If the cluster rounding dynamics can indeed be described by a simple physical model balancing surface tension and viscous dissipation, the same timescale should emerge for different clusters when normalized by size, and a master curve (data collapse) should be seen for the plot in Fig. 3d. In other words, the surface tension and viscosity as material parameters should be similar for different clusters.

We have added additional events to the rounding-up analysis. Fig. 3 (p. 8) in the revised manuscript is now based on the rounding up of 7 clusters. As the reviewer suggests, the data for all clusters lie within a narrow envelope, leading to a single best-estimate value of the ratio of surface tension to viscosity (γ/η).

4. The same comment as the above applies to the cluster fusion data in Fig. 4. Is it possible to get more statistics than just three events?

Unfortunately, we were unable to document more than three complete cluster fusion events due to multiple experimental limitations. The first limitation stems from the fact that it is impossible to predict *a priori* if two clusters that are physically close to one another will end up fusing or drifting apart. A second limitation is attributable to the highly dynamic nature of the clusters, so some fusion events only stay within the field of view for a portion of the entire fusion process. Finally, we targeted fusion events involving clusters of similar size, and this limited the number of events that could be monitored. If the reviewer deems that conclusions based on only three fusion events are not sufficiently reliable, we would propose to remove the cluster fusion section from the manuscript.

5. The fused cluster should also relax to a round shape because of surface tension. Do the authors observe this? If not, why not? And if yes, then can they quantify this rounding up in the same way as done for Fig. 3?

The reviewer raises a very interesting point. Indeed, in the viscoelastic liquid model that we propose here, the cluster should eventually relax to a round shape, as has been described previously (see cited Douezan *et al.* reference). This is indeed what we have observed where the fused cluster ultimately does relax to a round shape with similar dynamics as those observed during the rounding-up process. This information has been added to Supplementary Fig. S7, and corresponding text has been added to both Results (pp. 9-10) and Discussion (p. 15). We note, however, that a recent study has described cell aggregate fusion which does not relax to a round shape but rather attains an ellipsoidal equilibrium shape, a phenomenon known as "arrested coalescence" (Oriola *et al.*, "Arrested coalescence of multicellular aggregates", preprint, arXiv :2012.01455 [cond-mat.soft]). Arrested coalescence can be described by a viscoelastic solid model, rather than a viscoelastic liquid. Whether a cellular aggregate behaves as a viscoelastic liquid or a viscoelastic solid may depend on the presence and production of extracellular matrix or, as Oriola *et al.* propose, on the forces developed by contractile cell protrusions.

6. The authors should quantify cluster size distribution, and rate of growth of mean cluster size with time, and see if this leads to a power law expected for surface tension driven growth.

As had been shown in Supplementary Fig. S8 of the current version and reproduced below, cluster size is driven by competing factors including cluster growth and fusion on the one hand (which increase cluster area) and cluster rounding (which decreases the area) on the other hand. In our experiments, all these factors occur simultaneously because our recordings encompass the entire sequence of events. Therefore, the evolution of cluster size with time is not expected to simply follow surface tension driven growth. As depicted in the figure below for the low-, medium, and high-density clusters, we have quantified the rate of growth of mean cluster size and have determined that it indeed follows more complex behavior than a simple power law.

7. *The nematic order data in Fig. 6 is along expected lines, and doesn't have any particularly new message. Does the nematic orientation axis influence hole formation in any way? Have the authors looked for topological defects in the nematic order? These have been found to influence cell mounding or extrusion in layers of motile cells and could be interesting if found relevant in adherent cells as well.*

We agree with the reviewer that a potential correlation between topological defects in the nematic order and the occurrence of hole formation would be interesting. We were unable to investigate such a link, however, because cellular alignment in our multi-layer smooth muscle cell sheets was impossible to discern based on the live-cell brightfield images (see supplementary video "Movie 2: Dynamics of the opening of hole"). The nematic order data were obtained using fluorescence staining images of actin stress fiber orientation in fixed cells prior to cluster formation, and no information on where holes form in these fields is available. We have modified Fig. 6 (p. 13) in the revised manuscript to include a schematic of the hypothesized mechanism for cluster formation.

8. *Can the authors explain the radial organization of cells near the periphery of the cluster and circumferential along the central hole? Is this because of the differences in adhesion/anchoring that the cells have with the substrate vs a free interface?*

We believe that the reviewer's interpretation is correct as indicated in the first paragraph of the Results section (pp. 3-4). In further support of this notion, we have added supplementary information (Supplementary Fig. S1) showing that in addition to F-actin, smooth muscle actin (SMA), myosin heavy chain (MHC), and extracellular matrix (ECM; laminin and collagen IV) proteins are all prominently expressed along the periphery of the cluster. In the central zone, F-actin, SMA, MHC, and ECM proteins are expressed at much lower levels.

9. *The biological scope of the discussion seems a little narrow, given the emphasis on physical modeling. Similar dewetting/rounding up processes can also occur for single cells (such as cells just before dividing) and hole formation or closure could be related to wound healing or morphogenetic processes. Could the physical mechanisms identified by the authors apply to those situations, which also utilize cellular contractility and adhesion, even if the cell type is different? A discussion of this point could be interesting.*

We thank the reviewer for the suggestion. We have attempted to somewhat broaden the "biological scope" portion of the Discussion section while attempting to not resort to too much speculation. Moreover, we have noted the similarity in cellular organization between the clusters described here and that associated with certain features of various pathologies (p. 16).

Minor:

1. *In the panel for Fig. 2, the subfigures are not marked "a", "b", "c".... which makes reading the captions difficult. Also, I looked but couldn't find the "white arrow".*

We have made the requested modifications.

2. *A similar analysis of cytoskeletal filament clusters as liquid droplets which relax to characteristic shapes and fuse/coalescence was performed in Weirich et al. PNAS 2017 which the authors may find relevant. (<https://www.pnas.org/content/early/2017/02/09/1616133114>)*

We thank the reviewer for this information. We have added the reference (ref. 47) to the revised manuscript.

Reviewer #2

The manuscript by Wang et al provides a description of the formation of 3D clusters of SMCs from a monolayer. This experimental work is supported by a semi-quantitative theoretical analysis. The initiation of

the 2D to 3D transition is described as a brittle fracture and the growth of the 3D cluster as a dewetting process. This work is elegant by its simplicity. The authors only used timelapse video-microscopy, and they performed PIV and a few stainings on fixed samples for characterization. Still, they could propose an overall understanding of the process.

*By itself, the analogy between 3D cell aggregate formation from a cell sheet and the process of liquid dewetting has already been reported several times. Unfortunately, while the authors keep on referring to papers from previous works from one author (DGR) and other colleagues, they even do not refer to the proper paper (line 162, the dewetting paper from the Brochard-Wyart team to cite is not 28 or 29, but Douezan, S., Brochard-Wyart, F. Dewetting of cellular monolayers. *Eur. Phys. J. E* 35, 34 (2012)). The tendency to self-citation by the authors is highly regrettable. **I would recommend the authors to do a proper bibliographic search and give credit to other groups.** As an example, they could cite the work from the Trepat team (Pérez-González, C., Alert, R., Blanch-Mercader, C. et al. Active wetting of epithelial tissues. *Nature Phys* 15, 79–88 (2019). <https://doi.org/10.1038/s41567-018-0279-5>).*

We thank the reviewer for the information and have cited the Perez-Gonzalez *et al.* work in the revised manuscript (ref. 22).

In the end, my two major concerns with this manuscript are the following ones:

1. The paper lacks statistical analysis. One has the feeling that this phenomenon has been seen by chance and/or could be recorded properly only a couple of times. From that, the authors adapted an existing soft matter approach.

The reported phenomenon has actually been observed multiple times, and the results are quite robust. The revised manuscript includes additional information on the number of experiments and associated error bars and statistical analyses.

*Moreover, even though the authors propose that hole opening is controlled by elastic recoil and friction with the substrate, the estimate of the parameters is very rough. For instance, it seems that **they take the Young's modulus of individual cells instead of the one of cell sheet (that could be measured).***

The reviewer's remark is pertinent. We have modified the manuscript to account for the fact that the Young's modulus of a cell sheet is expected to be lower than that of a single cell (pp. 5-6). For keratinocytes, Vedula *et al.* (*Nature Materials* 6, 6111, 2015) reported a Young's modulus E for an epithelial cell sheet of ~ 2 kPa, versus reported literature values of E for single epithelial cells that are about 5 times larger. Assuming some generality to this result, we estimate the Young's modulus of the SMC layer to be of the order of 2 kPa, i.e. 5 times smaller than the single SMC Young's modulus of 10 kPa reported in the literature. This new estimate of the cell sheet elastic modulus yields an improved estimate of the cell-substrate friction coefficient, as discussed below.

*Founding the validity of the model on a **derivation of the friction coefficient, which is more than one order of magnitude larger than previous reports.***

We would first like to emphasize that the friction coefficient is used to model the dynamics of fracture opening only and is not present in the other model components (fracture shape, cluster shape and fusion, cluster population evolution). Regarding the fracture opening model, the reviewer is correct in that we estimated a value of the friction coefficient that was different from the estimates obtained by other authors for different experimental setups. We were not so much concerned with the exact value of the coefficient but with the physical picture. Keeping our original physical picture, the estimate can be refined by realizing that the contractile elastic force depends on the cross-section of the cell sheet, of thickness h , and thus scales with $R_c \cdot h$ rather than R_c^2 . Moreover, we have corrected our estimate of the Young's modulus E , which should be that of the cell sheet and not that of a single cell, as pointed out by the referee and addressed above. In the revised version of the manuscript we have introduced these two improvements, which yield a revised value

of the friction coefficient of $k=10^9$ Pa.s.m⁻¹, of the same order of magnitude as that of previous reports. This new estimate has been included in the revised manuscript (page 6 after Eq. 1).

*In other words, I think that, due to a lack of statistics and critical tests, the theoretical understanding of the process lies on a soft ground. In particular, **I cannot figure out why contractility impairment (using blebbistatin or soft substrate) only introduces a lag in the process instead of inhibiting it.***

As shown in Supplementary Fig. S9, blebbistatin at a concentration of 10 μ M actually inhibits cluster formation and does not simply slow down the process.

*Cluster rounding and coalescence of clusters should even not show up here, since these phenomena are known for ages. In particular, **the droplet model for spheroid coalescence used by the authors is quite debated** (see eg. Kosheleva, N.V., Efremov, Y.M., Shavkuta, B.S. et al. Cell spheroid fusion: beyond liquid drops model. *Sci Rep* 10, 12614 (2020). <https://doi.org/10.1038/s41598-020-69540-8>). It could have been of interest to test a more elaborate model by considering the aggregates as a viscoelastic material instead of a viscous liquid, as recently proposed (Dechristé G, Fehrenbach J, Grisetti E, Lobjois V, Poignard C. Viscoelastic modeling of the fusion of multicellular tumor spheroids in growth phase. *J Theor Biol.* 2018 Oct 7;454:102-109. doi: 10.1016/j.jtbi.2018.05.005).*

As the referee mentions, the liquid droplet model, while powerful in its simplicity, is not the only model that has been used to describe spheroid coalescence. The reviewer makes a valid point in that we should cite the literature more widely, and this issue has been addressed in the revised manuscript (first paragraph on p. 15). However, we disagree with the reviewer's assessment that cluster rounding and coalescence are phenomena that are no longer worth discussing. Indeed, to our knowledge, this is the first time these phenomena have been used to describe smooth muscle cell clusters.

In their paper, Kosheleva *et al.* present a satisfactory fit of the viscous liquid fusion model to their experiments. What they put into question is the meaning of the visco-capillary parameter in the model, which does not simply depend on cell type as in a naïve liquid analogy, but may also depend on the amount of extracellular matrix present in the aggregate or on cell migration characteristics. We should thus consider the fitted visco-capillary parameter as an "effective" value that accounts for the biological state of the system. We agree with this caveat. Indeed, the presence of extracellular matrix can explain why we obtain a smaller capillary velocity than that for rounding up: extracellular matrix is particularly present at the cluster periphery and thus slows down fusion while it has little effect on rounding. This point is discussed in the revised manuscript (p. 9), and the paper by Kosheleva is cited (ref. 37).

Regarding the paper by Dechristé *et al.*, the original element in their model is to consider the effect of cell proliferation during coalescence. In our experiments, cell proliferation occurs over a longer time scale than coalescence, so application of Dechristé *et al.*'s model is not pertinent. However, in order to broaden the bibliographic scope of our article, we now cite the paper of Dechristé *et al.* in the revised manuscript (ref. 36).

We also would like to clarify that the fusion model we use is appropriate for describing the long-time dynamics of viscoelastic drops. As has been shown in the literature by means of numerical simulations of viscoelastic behavior (see new references Hooper *et al.*, 2000(ref42), Scribber *et al.*, 2006(ref43)), viscoelastic sintering differs from purely viscous sintering over the short time scale (smaller than the viscoelastic time, of the order of minutes to hours), whereas the viscous approximation remains appropriate at the longer times reported here. This is why the viscous sintering model has been widely used to describe cell aggregate fusion by many groups (we have added several references to the paper to illustrate the widespread adoption of this model). The validity of the viscous approximation to describe fusion of a viscoelastic material is now discussed in the revised manuscript.

2. My second major criticism is that this work is sold as a work that unravels the underlying mechanisms of SMC-related diseases. This is the main focus in the abstract and in the introduction of the manuscript. To me,

the present work is primarily a nice soft matter-inspired biophysical work that suffers from a lack of statistics, sensitivity discussion and thorough critical tests.

With all due respect to the reviewer, we do not claim to “unravel the underlying mechanisms of SMC-related diseases”. We simply mention some diseases that involve abnormalities in smooth muscle cell organization as a motivation for the present study. We also discuss the potential implications of our results to some of these diseases. If the reviewer deems it necessary for us to further temper some of the wording, we will be happy to do so.

In conclusion, even though there is apparently nothing wrong in this paper, modeling is on a soft ground because several parameters are unknown and not directly measured, novelty is by far not obvious, and the biological motivation is a pretext of this “biological physics” work. I would therefore recommend publication in a more specialized soft matter journal

We hope that the extensive modifications made to the manuscript address the reviewers’ concerns including issues of statistics, parameter estimation, and biological relevance.

Reviewer #3

X. Wang and coworkers report a study on the spontaneous 2D to 3D transition of a layer of smooth muscle cells. They focus on the physics of this transition and quantitatively explain it, by invoking the sudden symmetry breaking (layer fracture) and the following dewetting.

In my opinion, this is a robust biophysical paper, which includes in vitro experiments, data analysis and modeling. However, I have noted some weaknesses in the article, which deserve to be reviewed.

Major points:

1. From the text, it is not clear how many times the experiments were reproduced. Figures 2, 3, 4 and 6 show single examples of data, and the corresponding fits. However, they do not provide a summary panel, reporting the fitting values obtained from different experiments and the distribution of such values.

Information on the number of experiments conducted in each part of the study has been added to the figures. All fits were conducted on the mean values.

2. Through the text, most of the results are given without errors (or standard deviations) and the number of samples. This information must be added.

We apologize for not having included complete information on the number of experiments and error ranges on which the conclusions are based in the original submission. This information has now been added to the figures of the revised manuscript.

3. A precise statement on reproducibility should be added. In particular, how many experiments have been done, how many succeeded, how many failed or gave different results (and why).

The number of experiments for each part of the study has now been added to the corresponding figure legends. Regarding experimental reproducibility more broadly, the phenomenon of cluster formation is highly reproducible using the methods described in the manuscript and provided that arterial (in our case aortic) smooth muscle cells are used. We note that in limited experiments, we tried similar experiments using human umbilical vein smooth muscle cells and observed that they failed to form clusters, consistent with the hypothesis of a contractility-driven mechanism. We have added this information to the Discussion section of the revised manuscript (bottom of p. 15 and top of p. 16).

4. In movie #1 something weird happens between $t = 31h00'$ and $t = 31h20'$. It seems that two movies have been merged, as there are no common features between the two frames. In addition, but also brightness and contrast suddenly change. I have hypothesized that the experimentalist changed some parameters after the fracture, such as the focus and the exposure time. Nevertheless, it is necessary to replace this movie with a better one, or – at least – provide a satisfying explanation of this sudden change.

This movie consists indeed of two segments that have been merged. They represent the exact same field of view with an interruption of 10-20 min due to a technical glitch during the recording. We have added a second movie of hole opening (Supplementary Movie 8), but this movie corresponds to a field with a less homogeneous cell sheet.

Minor points:

1. The authors insist on the fact that a low-density area appears in the center of the clusters shown in fig. 1a. In the text, they should explain better why this feature looks so interesting.

The reason we thought this organization was interesting is because it reminded us of some pathological settings such as atherosclerotic plaques and certain tumors that exhibit lower cell density in the core area. This information has been added to the Discussion (p. 16).

2. Movie 6: it seems that after formation the aggregate keeps attracting more and more cells from the subjacent layer. Is it true?

Movie 6 represents the stabilization phase of the cluster. During this phase, the round cluster continues to have peripheral attachment to the surrounding substrate. The anchoring cells are highly elongated due to contraction and align radially. When the contraction force overcomes the adhesion force, the anchoring cells detach from the substrate and are pulled to the cluster. This leads to the impression that the cluster keeps attracting cells from the subjacent layer.

3. Movie 7: the aggregates fuse, but do not fully merge. Do they ultimately form a unique aggregate, or a kind of interface remains between the two? If the fully merge, how long does it take? What is the mechanism of cell rearrangement, motility or death/division?

We consider that the fusion phase begins with the physical contact between the two clusters and ends with the fused cluster that has an elliptical shape. This ellipse eventually evolves in a circular shape in a “rounding-up” phase that resembles that described for a single cluster. To address one of the comments raised by reviewer #1, we have added an analysis of the “post-fusion rounding-up” (Supplementary Fig. S7 and Movie 9: post-fusion rounding-up) with corresponding text in both Results (pp. 9-10) and Discussion (p. 15). The post-fusion rounding-up is characterized by approximately the same velocity as the single cluster rounding-up. The precise mechanism of cellular rearrangement during fusion remains unknown at this point and certainly merits future investigation.

4. Figure 4c: check the unit: μm or μm^2 ?

We thank the reviewer for catching this error. The units should indeed be μm . The error has been corrected.

Reviewers' comments:

Reviewer #1 (Remarks to the Author):

In their revised version of this manuscript, the authors Wang et al. have addressed most of my previous concerns about the analyses and explanations. In my opinion, the submission is now much improved and has clearly presented the statistical information that all reviewers were concerned about. The observation of smooth muscle cells forming 3D clusters after first fracturing of the cell sheet, and physical understanding in terms of the scaling theory of soft materials, certainly makes this a strong paper. My few remaining concerns and comments in the order of decreasing importance are as follows:

1. In response to my previous comment #7 on measuring the nematic order of actin fibers, the authors explain that nematic order was not possible to measure in the live cell sheet that undergoes fracturing and hole formation. While their point is well taken, I believe the nematic order analysis can be more informative than what is reported in Fig. 6b. It appears as if the authors have only measured the global nematic order parameter, but they might also want to measure the local nematic order parameter (e.g., by dividing their image into smaller boxes) to obtain a spatial map of the nematic order. Such an analysis can reveal if the nematic order is homogeneous or if there are local regions of depleted order. This latter would suggest defects. If the order is homogeneous, it is still worth mentioning in the SI.

2. On page 8, the role of substrate stiffness is briefly discussed. Presumably the slower dynamics of cluster formation is because of the lower contractile forces exerted by the cells. However, have the authors done this analysis with the cluster fusion data on the soft substrate. Although the cell contractility is lower, the substrate is also easier to remodel when two clusters fuse, and this should lead to a compensating effect.

3. In Fig. 3d, it would help to see the data points from different clusters plotted as different symbols to see how much spread there really is in the data. Each dataset can be fit separately to Eq. 3 to see how much spread there is in the time scale.

4. In the caption to Fig. 3c, it would help to see how "cluster circularity" is defined, probably in terms of the variables c and a defined below Eq. 3.

5. The typesetting of Equations could be better. For example the "C" in Eq. 1 and also the * symbols for multiplication look a little strange, at least in my copy.

Reviewer #2 (Remarks to the Author):

The authors have significantly revised the manuscript and clarified some points that were obscure in the initial version. In particular, as recommended by the reviewers, they now picked more reasonable values for the parameters of the young's modulus of a cell monolayer. This allowed them to find friction coefficients that are finally in the range of those of previous reports, which is an indirect proof that their theoretical model makes sense.

Concerning the specific answer to my comments, I regret that they did simply wipe the question about that blebbistatin effect and contractility impairment. By writing that « I cannot figure out why... », I was implicitly expecting some explanation and interpretation/speculation from the authors. They just confirmed, without any attempt of mechanistic interpretation. So, I rephrase my question : Could the authors explain why blebbistatin does not inhibit but simply slows down the process ?

Finally, and more importantly, all referees noted the lack of statistics. The authors answered that 'The reported phenomenon has actually been observed multiple times... » . Hopefully !, « ...and the results are quite robust... « . The « robustness of results is not a feeling but needs to be quantified through

statistical analysis. Altogether, hole formation and rounding processes are claimed to be derived from 5 and 7 experiments respectively. I would be very interested in having access to all the raw data. Indeed, I don't understand why the mean and SD closely follow the « accidents », which may be due to loss of focus or substrate heterogeneity (see the slight shift in the normalized area in Fig 3b, or the bump in Figure 3d, between 150 and 200h). I cannot imagine such a reproducibility between all experiments. A full access to the raw data and explanation of how these data were further exploited for model fitting seems to be critical. A model cannot be validated on one occurrence in a high impact journal like *Commbio*. I could only suggest that the authors perform new experiments. But, I am still not quite convinced that the biological relevance is sufficient for publication in *Commbio*, especially if they « further temper the wording » about the biological/medical implications. A specialized soft matter physics journal would be by far more appropriate.

Reviewer #3 (Remarks to the Author):

In the resubmitted version, the authors addressed satisfactorily all my question and concerns.

Response to Reviewers

We would like to thank the reviewers for their latest comments to which we have endeavored to positively respond. Below is a point-by-point response to the specific points raised by the referees. The reviewers' comments are shown in italics.

Reviewer #1 (Remarks to the Author):

In their revised version of this manuscript, the authors Wang et al. have addressed most of my previous concerns about the analyses and explanations. In my opinion, the submission is now much improved and has clearly presented the statistical information that all reviewers were concerned about. The observation of smooth muscle cells forming 3D clusters after first fracturing of the cell sheet, and physical understanding in terms of the scaling theory of soft materials, certainly makes this a strong paper. My few remaining concerns and comments in the order of decreasing importance are as follows:

We thank the reviewer for the positive comments on the revised manuscript.

1. In response to my previous comment #7 on measuring the nematic order of actin fibers, the authors explain that nematic order was not possible to measure in the live cell sheet that undergoes fracturing and hole formation. While their point is well taken, I believe the nematic order analysis can be more informative than what is reported in Fig. 6b. It appears as if the authors have only measured the global nematic order parameter, but they might also want to measure the local nematic order parameter (e.g., by dividing their image into smaller boxes) to obtain a spatial map of the nematic order. Such an analysis can reveal if the nematic order is homogeneous or if there are local regions of depleted order. This latter would suggest defects. If the order is homogeneous, it is still worth mentioning in the SI.

We have followed the reviewer's suggestion of zooming in on a more local scale in the actin staining images and have investigated the possibility of occurrence of topological defects. As depicted in the three examples below, topological defects can indeed be discerned in a number of cases, thereby indicating the presence of local regions of depleted order. As explained in our response in the previous review cycle, we are unable to link these topological defects which are obtained from images of fixed cells with what occurs in the live cell sheet during the process of monolayer fracturing and subsequent hole formation. We hope this additional information addresses the reviewer's concern.

Figure 1. Examples of $-1/2$, $+1/2$, and $+1$ defects in the cell layer based on the orientation of actin stress fibers.

2. On page 8, the role of substrate stiffness is briefly discussed. Presumably the slower dynamics of cluster formation is because of the lower contractile forces exerted by the cells. However, have the authors done this analysis with the cluster fusion data on the soft substrate. Although the cell contractility is lower, the substrate is also easier to remodel when two clusters fuse, and this should lead to a compensating effect.

We were only able to observe a single cluster fusion event on the softer substrate. A comparison of the fusion dynamics on the hard substrates (HS) to those on the soft substrates (SS) is shown below. The fitting of the soft substrate data yielded a γ/η of 1.8×10^{-9} m/s, somewhat smaller than the value we had reported for the hard substrate (2.9×10^{-9} m/s).

3. In Fig. 3d, it would help to see the data points from different clusters plotted as different symbols to see how much spread there really is in the data. Each dataset can be fit separately to Eq. 3 to see how much spread there is in the time scale.

The plot requested by the reviewer is shown below. The table below shows the γ/η value obtained for each rounding event. As can be seen from the table, the γ/η values all lie between 6×10^{-9} m/s and 1.3×10^{-8} m/s. We hope this information addresses the reviewer's concern.

Rounding event	γ/η (m/s)
R1	1.3×10^{-8}
R2	6×10^{-9}
R3	8×10^{-9}
R4	1.1×10^{-8}
R5	6×10^{-9}
R6	1.1×10^{-8}
R7	6×10^{-9}

4. In the caption to Fig. 3c, it would help to see how "cluster circularity" is defined, probably in terms of the variables c and a defined below Eq. 3.

The definition of cluster circularity has been added to the caption of Fig. 3c. Cluster circularity (sometimes referred to as shape index in the literature) is defined as $4\pi A/P^2$, where A is the cluster area and P is its perimeter.

5. The typesetting of Equations could be better. For example the " C " in Eq. 1 and also the * symbols for multiplication look a little strange, at least in my copy.

We do not see an issue with the typesetting of Eq. 1 in our copy. We will ensure that the equations are all correctly typeset in the final version of the manuscript.

Reviewer #2 (Remarks to the Author):

The authors have significantly revised the manuscript and clarified some points that were obscure in the initial version. In particular, as recommended by the reviewers, they now picked more reasonable values for the parameters of the young's modulus of a cell monolayer. This allowed them to find friction coefficients that are finally in the range of those of previous reports, which is an indirect proof that their theoretical model makes sense.

We thank the reviewer for the positive comments on the revised manuscript.

Concerning the specific answer to my comments, I regret that they did simply wipe the question about that blebbistatin effect and contractility impairment. By writing that « I cannot figure out why... », I was implicitly expecting some explanation and interpretation/speculation from the authors. They just confirmed, without any attempt of mechanistic interpretation. So, I rephrase my question: Could the authors explain why blebbistatin does not inhibit but simply slows down the process?

As the reviewer undoubtedly knows, blebbistatin is a myosin II inhibitor, which certainly inhibits cellular contractility but also impacts the strength of cell adhesion to the substrate. The blebbistatin doses and incubation times used here were a compromise between a significant reduction in cell contractility and the need to maintain an adherent cell sheet. We believe that the significant slowdown in cluster formation upon blebbistatin treatment is supportive of the assertion that contractility plays an important role in this process.

Finally, and more importantly, all referees noted the lack of statistics.

This was indeed a concern expressed by all the reviewers of the original submission. In our revised manuscript, those concerns seem to have been addressed for the other two reviewers.

*The authors answered that ‘The reported phenomenon has actually been observed multiple times... ». Hopefully !, « ...and the results are quite robust.... « The « robustness of results is not a feeling but needs to be quantified through statistical analysis. Altogether, hole formation and rounding processes are claimed to be derived from 5 and 7 experiments respectively. I would be very interested in having access to all the raw data. Indeed, I don’t understand why the mean and SD closely follow the « accidents », which may be due to loss of focus or substrate heterogeneity (see the slight shift in the normalized area in Fig 3b, or the bump in Figure 3d, between 150 and 200h). I cannot imagine such a reproducibility between all experiments. A full access to the raw data and explanation of how these data were further exploited for model fitting seems to be critical. A model cannot be validated on one occurrence in a high impact journal like *Commbio*. I could only suggest that the authors perform new experiments.*

We have provided an Excel table that includes the raw data requested by the reviewer both for the values of $\varepsilon/\varepsilon_0$ used in the γ/η fitting and for hole width as a function of time.

*But, I am still not quite convinced that the biological relevance is sufficient for publication in *Commbio*, especially if they « further temper the wording » about the biological/medical implications. A specialized soft matter physics journal would be by far more appropriate.*

We respectfully disagree with the reviewer in this regard. As indicated in the text, cluster formation in high density smooth muscle cell sheets has been repeatedly reported over many years. We believe that the fact that we provide a quantitative description of this process that potentially has important physiological and pathological implications merits publication in a high impact journal such as *Communications Biology*.